# Prime editing with genuine Cas9 nickases minimizes unwanted indels

Jaesuk Lee [1,2,7], Kayeong Lim [1,3,7], Annie Kim[1], Young Geun Mok[1,6], Eugene Chung[1,2], Sung-Ik Cho [1,2,4], Ji Min Lee[1,2] & Jin-Soo Kim [1,5] ✉

Unlike CRISPR-Cas9 nucleases, which yield DNA double-strand breaks (DSBs), Cas9 nickases (nCas9s), which are created by replacing key catalytic amino-acid residues in one of the two nuclease domains of S. pyogenesis Cas9 (SpCas9), produce nicks or single-strand breaks. Two SpCas9 variants, namely, nCas9 (D10A) and nCas9 (H840A), which cleave target (guide RNA-pairing) and non-target DNA strands, respectively, are widely used for various purposes, including paired nicking, homology-directed repair, base editing, and prime editing. In an effort to define the off-target nicks caused by these nickases, we perform Digenome-seq, a method based on whole genome sequencing of genomic DNA treated with a nuclease or nickase of interest, and find that nCas9 (H840A) but not nCas9 (D10A) can cleave both strands, producing unwanted DSBs, albeit less efficiently than wild-type Cas9. To inactivate the HNH nuclease domain further, we incorporate additional mutations into nCas9 (H840A). Double-mutant nCas9 (H840A + N863A) does not exhibit the DSB-inducing behavior in vitro and, either alone or in fusion with the M-MLV reverse transcriptase (prime editor, PE2 or PE3), induces a lower frequency of unwanted indels, compared to nCas9 (H840A), caused by error-prone repair of DSBs. When incorporated into prime editor and used with engineered pegRNAs (ePE3), we find that the nCas9 variant (H840A + N854A) dramatically increases the frequency of correct edits, but not unwanted indels, yielding the highest purity of editing outcomes compared to nCas9 (H840A).

Cas9 nuclease creates double-strand breaks (DSBs) in target DNA, which enables genome engineering via non-homologous end joining (NHEJ) or homology directed repair (HDR) in eukaryotic cells[1–5]. However, DSBs are highly toxic lesions that can lead to unwanted genome rearrangements, including large deletions[6–8], and/or even cell death[9,10]. As an alternative to DSB-mediated genome engineering, base editing[11–13] and prime editing[14] systems, which involve Cas9 nickase (nCas9) rather than Cas9 nuclease, do not require DSBs. Alanine substitutions at D10 in the RuvC domain or H840 in the HNH domain of

Cas9 convert the nuclease into distinct nickases, which cleave the target and non-target DNA strand, respectively. Base editors such as cytosine base editor[11,12] and adenosine base editor[13] can be generated by fusing cytosine deaminase or adenosine deaminase to dead Cas9 (dCas9). To increase their editing efficiency, dCas9 is replaced with nCas9 (D10A). Unlike dCas9, nCas9 (D10A) nicking of the target strand stimulates cellular repair mechanisms, which leads to increased editing frequencies by both cytosine and adenosine base editors in the absence of a DSB.

[1]Center for Genome Engineering, Institute for Basic Science, Daejeon, Republic of Korea. [2]Department of Chemistry, Seoul National University, Seoul, Republic of Korea. [3]Brain Science Institute, Korea Institute of Science and Technology, Seoul, Republic of Korea. [4]Department of Pharmacology, Yonsei University College of Medicine, Seoul, Republic of Korea. [5]Department of Biochemistry and NUS Synthetic Biology for Clinical & Technological Innovation (SynCTI), National University of Singapore, Singapore, Singapore. [6]Present address: GreenGene Inc, Seoul, Republic of Korea. [7]These authors contributed equally: Jaesuk Lee, Kayeong Lim. ✉e-mail: jskim01@snu.ac.kr

On the other hand, prime editor (PE)[14], a programmable genome editing tool that enables nucleotide substitutions, insertions, and deletions, consists of nCas9 (H840A) and Moloney murine leukemia virus (M-MLV) reverse transcriptase (RT) domains. Prime editing systems are guided by prime editing guide RNAs (pegRNAs) that have an extended structure on the 3' end compared to single-guide RNAs (sgRNAs). This extension contains the primer binding site (PBS), which is complementary to a portion of the DNA protospacer, and a RT template that encodes the intended edit. Prime editing is initiated by PE binding to the target sequence and nicking of the non-target strand by nCas9 (H840A). Once nicking occurs, the PBS sequence pairs with the complementary target DNA sequence to start priming reverse transcription from the RT template, which enables the desired editing. However, unlike base editors, PE induces relatively high frequencies of unwanted indels[14–16], which reduces the purity of the desired product. In this study, we demonstrated that, in vitro, purified nCas9 (H840A) can also sometimes create DSBs. Additional mutagenesis in the HNH domain of nCas9 (H840A) removed this ability and reduced nCas9 (H840A)-mediated indel generation in HEK293T cells. Digenome-seq validated that this improved nCas9 variant could reduce the frequency of off-target, genome-wide DSBs. We reasoned that because PE includes a nCas9 (H840A) domain, a high proportion of unwanted, PE-associated indels may be the result of this ability to generate DSBs. Therefore, this improved version of nCas9 was incorporated into later generations of PEs (PE2, PE3, and PE3 with engineered pegRNAs (epegRNAs)) to minimize the unwanted indels.

## Results

### nCas9 (H840A) sometimes creates DSBs

Cas9 nuclease enables programmable genome engineering via NHEJ or HDR by creating DSBs at target sites. In contrast, more recently developed genome editing tools such as base editors and PEs use nCas9 (D10A) and nCas9 (H840A), which nick the target and non-target strands, respectively (Fig. 1a). To confirm the cleavage patterns generated by these enzymes, we treated supercoiled plasmid in vitro with purified Cas9, nCas9 (D10A), or nCas9 (H840A) proteins together with in vitro transcribed sgRNA targeted to the *HEK4* site. The nicking endonuclease Nt.BbvCI and the restriction enzyme SpeI (which generates DSBs) were used as controls. Nicking of supercoiled plasmids generates an open circular form, which exhibits an apparent increase in size on agarose gels compared to the supercoiled form, whereas linearized plasmids exhibit an apparent decrease in size (Fig. 1b). These size differences could potentially be used to describe the functional cleavage patterns of Cas9-related proteins. Digestion of supercoiled plasmids with Cas9 leads to almost complete linearization (99.7%), whereas treatment with nCas9 (D10A) generates primarily the open circular form (84.0%). However, in contrast to our expectations, two major products resulted from treatment with nCas9 (H840A): both open-circular (56.7%) and linearized forms (43.4%) (relative band intensities were calculated using ImageJ software) (Fig. 1c).

To examine the cleavage patterns generated by these enzymes further, we performed Digenome-seq[17–19]. Purified Cas9, nCas9 (D10A), and nCas9 (H840A) proteins, together with in vitro transcribed *HEK4*-targeting sgRNA, were incubated with genomic DNA isolated from HEK293T cells. Cleavage patterns of Cas9, nCas9 (D10A), and nCas9 (H840A) were examined by whole genome sequencing (WGS) and examined with the IGV viewer. The anticipated cleavage patterns of these enzymes at the on-target site are presented in Fig. 1d. Cas9 and nCas9 (D10A) caused the expected cleavage patterns. However, surprisingly, nCas9 (H840A) completely cleaved the non-target strand, and partially cleaved the target strand, resulting in unexpected DSBs (Fig. 1e and S1A–S1D).

### Addition of an N863A mutation to nCas9 (H840A) results in an enzyme that catalyzes single-strand breaks in the non-target strand

Our experiments so far have demonstrated that nCas9 (H840A) can sometimes generate DSBs. We reasoned that because target strand cleavage is catalyzed by the Cas9 HNH domain, the H840A mutation may not be enough to completely disable this HNH activity. Therefore, to create a complete non-target strand nickase Cas9, additional engineering in the catalytic region of the HNH domain may be needed to completely inactivate it. To test this hypothesis, we added another mutation to nCas9 (H840A). Structural studies of the Cas9 HNH domain indicated that residue N863 makes functional contact with H840[20]. N863 plays a role in coordinating with an $Mg^{2+}$ ion required for catalysis when SpCas9 is in the cleavage state (state II). Therefore, we added the N863A mutation to nCas9 (H840A) in an effort to eliminate the function of the HNH domain. Next, purified Cas9, nCas9 (H840A), and nCas9 (H840A + N863A) proteins, together with in vitro transcribed sgRNAs (targeting the *HEK4*, *EMX1*, and *RUX1* sites), were incubated with genomic DNA isolated from HEK293T cells. The cleavage patterns of these enzymes were then examined by WGS and visualized using the IGV viewer. As hypothesized, nCas9 (H840A + N863A) did not cleave the target strand, and, instead, generated clean single-strand breaks in the non-target strand at the *HEK4*, *EMX1*, and *RUNX1* sites (Fig. 2a).

Because nCas9 (H840A) sometimes creates DSBs at on-target sites, it would also be expected to generate DSBs at genome-wide off-target sites; likewise, given that nCas9 (H840A + N863A) does not generate DSBs at on-target sites, it could avoid off-target DSBs. To investigate the pattern of genome-wide DSBs following treatment with nCas9 (H840A) and nCas9 (H840A + N863A), WGS data were subjected to Digenome-seq, a method that captures genome-wide off-target loci based on the in vitro cleavage pattern. DSBs and other base changes including C-to-U and A-to-I created by genome editing tools such as Cas9 nucleases, base editors, and PEs can be detected by this method[17,21–24]. Genomic DNA samples treated with wild-type (WT) Cas9, nCas9 (H840A), or nCas9 (H840A + N863A) targeted to the *HEK4*, *RUNX1*, and *EMX1* sites were subjected to Digenome-seq and their captured genome-wide DSB sites are shown using Circos plots (on-target sites are indicated by black arrowheads; the height of the black bars represents the Digenome score) (Fig. 2b). The number of DSB sites captured in the Cas9-treated samples ranged from 148 (*EMX1*-targeted sgRNA) to 454 (*HEK4*-targeted sgRNA). In the nCas9 (H840A)-treated samples, 8 (*RUNX1*) to 23 (*HEK4*) DSB sites were captured. However, only a few DSB sites were detected in the nCas9 (H840A + N863A)-treated samples (in the range of 2 to 3 sites) in all cases (Fig. 2c). On average, $260 \pm 100$, $24 \pm 10$, $2.7 \pm 0.3$, and $2.0 \pm 0.0$ DSB sites were detected in the Cas9-, nCas9 (H840A)-, and nCas9 (H840A + N863A)-treated samples and the untreated control, respectively (Fig. 2d). In summary, the addition of the N863A mutation to nCas9 (H840A) almost completely eliminates the ability of this enzyme to create DSBs at both on-and off-target sites.

To examine whether off-target indels were induced by nCas9 (H840A), nCas9 (H840A + N863A), and WT Cas9, we examined indel frequencies in *RUNX1*- and *EMX1*-targeted samples at candidate off-target sites. DSB sites detected by nCas9 (H840A)-induced Digenome-seq (Fig. 2b) were used for validation. The Digenome score ($\geq 8.0$) and the number of mismatched bases ($\leq 6$ bp) were applied to filter candidate sites. Indel frequencies induced by nCas9 variants and WT Cas9 at off-target sites in the *RUNX1*- (Fig. S2A) and *EMX1*- (Fig. S2D) targeted samples were then measured. In addition, because prime editor uses nCas9 (H840A), we also examined whether off-target indels were generated by PE2 and PE2 (H840A + N863A). The same off-target candidate sites were tested for intended edits and unwanted indels in *RUNX1*- (Fig. S2B and S2C) and *EMX1*- (Fig. S2E and S2F) targeted samples. We

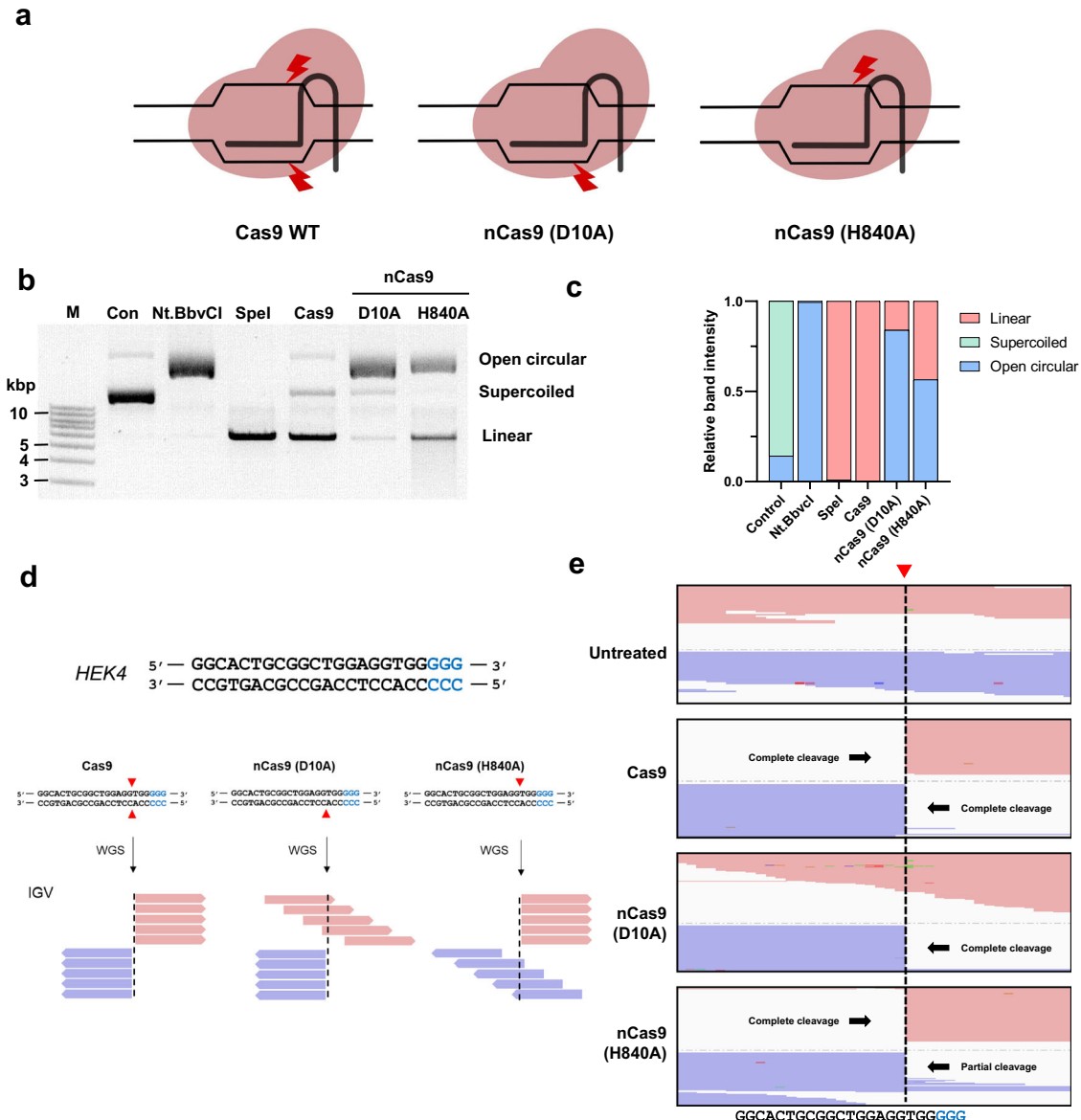

**Fig. 1 | nCas9 (H840A) can create DSBs. a** A schematic diagram of the anticipated cleavage patterns of WT Cas9, nCas9 (D10A), and nCas9 (H840A). **b** In vitro plasmid digestion assay. Supercoiled plasmids were digested with nicking endonuclease Nt.BbvCI, restriction enzyme SpeI, and purified WT Cas9, nCas9 (D10A), and nCas9 (H840A) proteins. Nicks or DSBs in supercoiled plasmids generate open-circular or linear forms, respectively. **c** Relative band intensities of the supercoiled, open circular, and linear forms were calculated using ImageJ software. **d** Expected cleavage patterns at the *HEK4* target site as visualized by the IGV viewer. Red: forward strand; blue: reverse strand; arrow head: cleavage point; blue characters: protospacer adjacent motif (PAM). **e** The actual cleavage patterns after in vitro digestion of genomic DNA with purified WT Cas9, nCas9 (D10A), and nCas9 (H840A) proteins together with *HEK4*-targeting sgRNA. Gel image (**b**) represents one of three independent experiments with similar results. Source data are provided as a Source Data file.

were able to confirm the presence of low frequencies of indels induced by nCas9 (H840A) and PE2 at some off-target sites. Average indel frequencies of 0.027% and 0.035% were detected at the RUNX1 off-2 site in nCas9 (H840A)- and PE2-treated samples, respectively.

### Additional mutations in the HNH domain further reduce indel formation

To engineer a Cas9 nickase that would only cleave the non-target strand, we mutated additional residues in the Cas9 HNH domain that have been implicated in target strand cleavage. Using the structure of SpCas9 in cleavage state II (PDB:6O0Y), amino acids within 5 Å of H840 were chosen for further engineering[25] (Fig. S3A). N854 and D839, in the HNH domain were selected and mutated to alanine (Fig. S3B). We generated 14 different versions of nCas9, containing different combinations of the D839A, H840A, N854A, and N863A mutations.

We reasoned that if the HNH domain were made completely dysfunctional by additional mutagenesis, indel generation in HEK293T cells treated with the engineered Cas9 enzymes would decrease. To test this idea, we delivered plasmids encoding WT Cas9 or nCas9 variants and sgRNAs targeting 15 endogenous genomic sites into HEK293T cells. Then, indel frequencies induced by Cas9 and the nCas9 variants were measured by targeted deep sequencing. We found that indel frequencies induced by Cas9 ranged from 24% to 80% (on average, 63 ± 2%), whereas frequencies induced by nCas9 (H840A) ranged from 0.050% to 15% (on average, 2.5 ± 0.6%), at the 15 endogenous sites. Interestingly, as we expected, nCas9 (H840A + N863A) induced a significantly lower average indel frequency of 0.34 ± 0.06% at the 15 target sites. As we had hypothesized, we identified versions of nCas9 with additional mutations that induced even lower indel frequencies than nCas9 (H840A + N863A). Three nCas9 variants

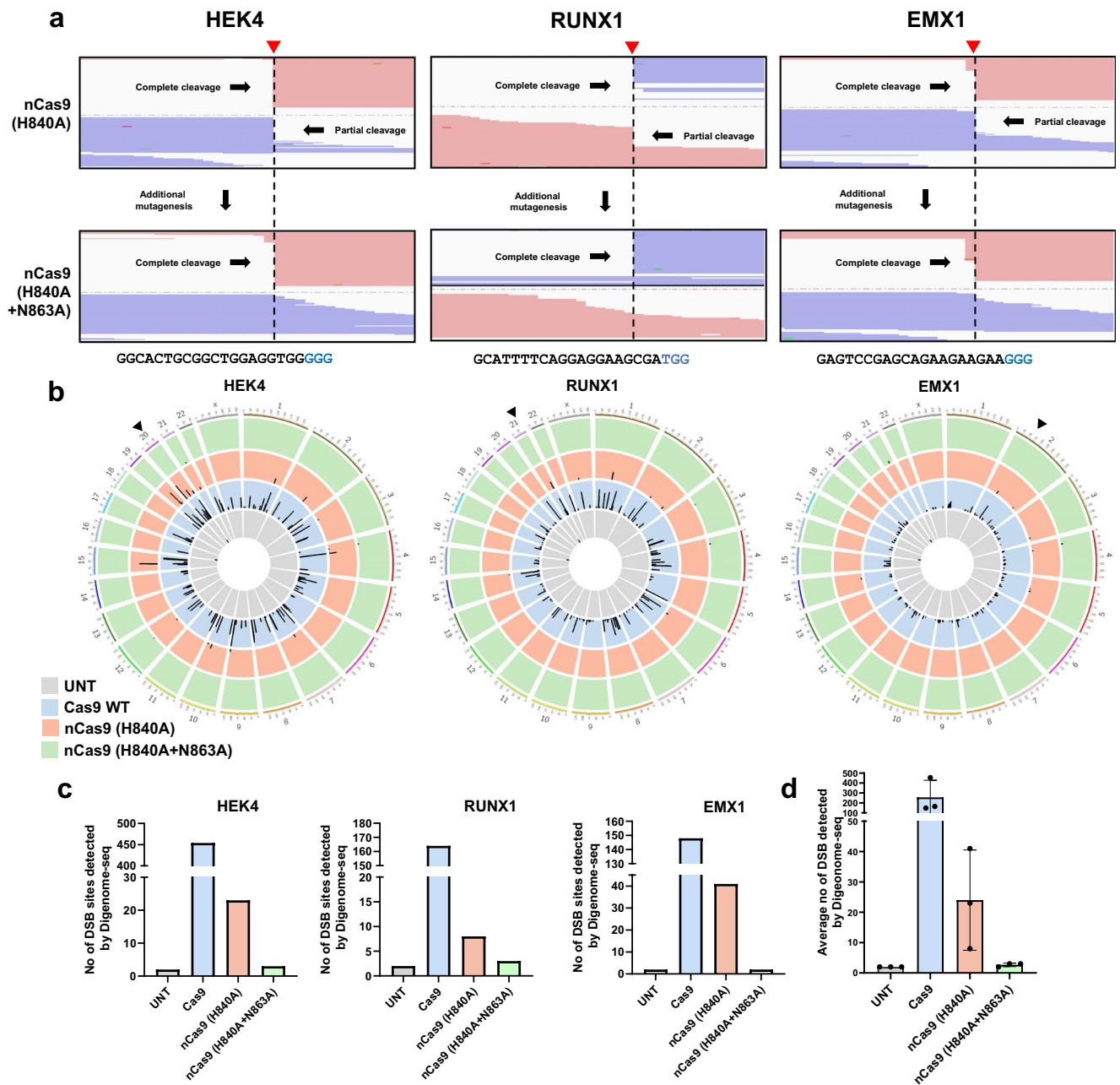

**Fig. 2 | Target strand cleavage catalyzed by nCas9 (H840A) can be reduced by an additional mutation at N863 in the HNH domain. a** WGS results at on-target sites visualized with the IGV viewer. Genomic DNA isolated from HEK293T cells was subjected to in vitro digestion with purified nCas9 (H840A) and nCas9 (H840A + N863A) proteins with appropriate sgRNAs. The addition of an N863A mutation to nCas9 (H840A) removed its target strand-cleaving function, as seen at three different loci. **b** Circos plots showing genome-wide DSBs detected by Digenome-seq. Genome-wide DSB sites created by purified WT Cas9, nCas9 (H840A), and nCas9 (H840A + N863A) proteins were captured. Arrow heads: on-target sites; height of black bars: Digenome scores. **c** Number of DSB sites found by Digenome-seq at three loci. **d** Average numbers of genome-wide DSBs found in samples treated with WT Cas9, nCas9 (H840A), and nCas9 (H840A + N863A). Mean ± SEM (**d**) was determined three independent experiments. Source data are provided as a Source Data file.

(H840A + N854A, H840A + N863A + N854A, and H840A + N863A + D839A + N854A) induced indel frequencies of 0.03 ± 0.01%, 0.02 ± 0.01%, and 0.03 ± 0.01%, respectively, which represent significant reductions compared to that of nCas9 (H840A + N863A) (Fig. S3C). Thus, additional mutations affecting catalytic amino acids in the HNH domain could further reduce indel generation in HEK293T cells. However, these additional mutations in the HNH domain may also reduce its general functionality, such as its binding activity or protein folding, which may in turn reduce the efficiency of indel generation. To determine whether these nCas9 variants are still functional, we have incorporated them into the prime editor system.

## PE2 variants containing improved versions of nCas9 induce fewer unwanted indels

Relatively high frequencies of unwanted indels are one of the problems associated with prime editing. Because PE includes nCas9 (H840A), we reasoned that the ability of nCas9 (H840A) to generate DSBs could be the source of this problem. Therefore, we incorporated our engineered nCas9 variants into PE2 to determine whether they could reduce the rate of unwanted indels.

We assessed the activity of these PE2 systems, programmed to install single-base mutations, at 12 target sites in HEK293T cells (Fig. 3a). The frequency of intended base edits induced by PE2 (H840A)

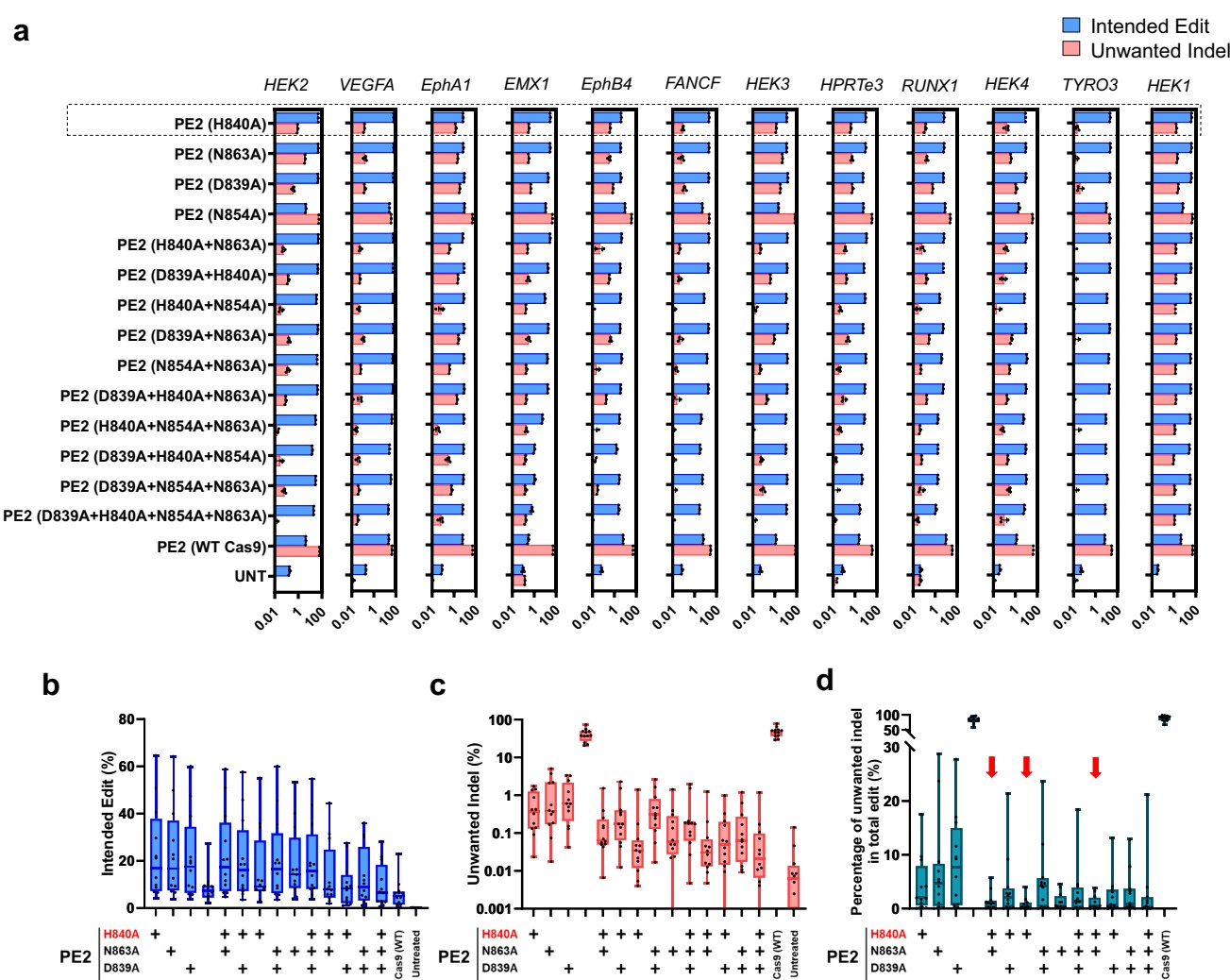

**Fig. 3 | PE2 variants that incorporate improved nCas9 variants induce a reduced frequency of unwanted indels. a** Plasmids encoding PE (nCas9 variants) and individual pegRNAs targeting 12 endogenous loci were transfected into HEK293T cells. **b**, **c** The frequencies of intended edits and unwanted indels at each site are presented in the graphs. Average frequencies of intended edits (**b**) and unwanted indels (**c**). **d** The percentage of unwanted indels among all edited sequences (unwanted indels + intended edits) for the 12 target sites. The red arrows

indicate the three variants that reduce the frequency of unwanted indels the most, relative to all edited sequences. Mean ± SEM (**a**) were determined three independent experiments. Mean ± SEM (**b**–**d**) of all individual values of sets of $n = 3$ independent replicates are shown. Data of minimum-to-maximum values are presented. For the boxes (**b**–**d**), the top, middle, and bottom lines represent the 25th, 50th, and 75th percentiles, respectively. The whiskers indicate min to max values. Source data are provided as a Source Data file.

was 23 ± 6% on average (ranging from 4.2% to 65%). PE2 systems containing nCas9 with N863A, D839A, H840A + N863A, D839A + H840A, D839A + N863A, N854 A + N863A, and D839A + H840A + N863A mutations in the HNH domain retained the desired single-base editing activity (19% to 23%, on average), which is comparable to that of PE2 (H840A). This finding shows that PE2 systems incorporating these HNH variants cleaved the non-target strand, essential for successful prime editing (Fig. 3b). Even the PE containing Cas9 nuclease exhibited an intended single-base editing activity of 6.1 ± 1.7% (ranging from 0.30% to 23%), although it also induced the highest frequency of indels (48 ± 4%). PE2 systems involving all other tested HNH domain mutations displayed intended editing efficiencies that were reduced by more than 80% compared to that of the conventional PE2 (H840A).

We then examined the frequency of unwanted indels. PE2 (H840A) induced unwanted indels at a frequency of 0.60 ± 0.17% (ranging from 0.023% to 1.7%). PE2 systems containing H840A + N863A, H840A + N854A, H840A + N854A + N863A, D839A + H840A + N854A, and D839A + H840A + N854A + N863A mutations reduced the frequency of unwanted indels by 2.6-, 3.8-, 4.2-, 3.8-, and 4.4-fold, respectively, compared to that induced by PE2 (H840A) (Fig. 3c). To

select the PE2 variants that retained the ability to induce intended edits efficiently while generating fewer unwanted indels, we calculated the ratio of unwanted indels to total edits (intended edits + unwanted indels) in each case. The average ratio of unwanted indels for PE2 (H840A) at 12 endogenous target sites was 4.3%. Notably, we found that PE2 (H840A + N863A), PE2 (H840A + N854A), and PE2 (H840A + N864A + N863A) were associated with unwanted indel ratios of only 1.1%, 0.77% and 1.1% respectively (Fig. 3d). Thus, our improved versions of nCas9 incorporated into the PE2 system can reduce the ratio of unwanted indels, while maintaining intended editing outcomes.

Additionally, given that a deletion of the HNH domain of Cas9 is tolerated for its DNA binding function[26], we constructed and tested HNH-deleted nCas9 variants. Four different fragments representing all or part of the HNH domain (residues 792-897, 765-908, 786-885, 824-874) were deleted and replaced with variable linkers of 2, 5, or 10 amino-acid residues in length (Fig. S4A). The resulting HNH-deleted nCas9 variants were then incorporated into PE2. We measured the frequencies of intended edits and unwanted indels induced by these PE2 variants targeted to 12 endogenous sites in HEK293T cells (Fig. S4B). The Δ792-897 and Δ786-885 variants exhibited intended

editing efficiencies about half that of PE2 (H840A), but all 12 HNH-deleted variants reduced the frequencies of both intended editing and unwanted indels (Fig. S4C–E). Based on our data from PE2 variants containing HNH-substituted and HNH-deleted nCas9 variants, we focused on the three variants that resulted in the highest editing purity (H840A + N863A, H840A + N854A, and H840A + N864A + N863A) for further investigation (Fig. 3d).

## PE variants containing nCas9 variants reduce the frequency of unwanted indels

Next, we tested the selected nCas9 variants in the PE3 system: because PE2 exhibits a relatively low efficiency of genome editing, an additional guide RNA, namely a nicking sgRNA, is used to induce nicking in the opposite strand of DNA to increase the editing efficiency by stimulating cellular repair mechanisms. However, because the PE3 system uses two guide RNAs (pegRNA and sgRNA), PE3 (H840A) could generate DSBs at two sites via the activity of nCas9, increasing the yield of unwanted indels. Therefore, an appropriate nCas9 variant incorporated into PE3 should also reduce the frequency of unwanted indels induced by this system. To test our hypothesis, we examined the effect of using nCas9 variants in the PE3 system programmed to install five different single-base substitutions at three target sites (HEK3, RUNX1, and FANCF) (Fig. 4a–c). The frequency of intended edits induced by PE3 (H840A) ranged from 7.6% to 51% (on average, 32 ± 2%). Importantly, when PE3 (H840A + N863A) was used instead, the average frequency of correct edits was not significantly different that induced by PE3 (H840A) (on average, 30 ± 2%), but the average frequency of unwanted indels dropped significantly, from 4.3 ± 0.4% for PE3 (H840A) to 2.6 ± 0.3% for PE3 (H840A + N863A) (Fig. 4d). PE3 (H840A + N854A) and PE3 (H840A + N854A + N863A) induced low frequencies of unwanted indels, but also induced low frequencies of intended edits (average, 9.9 ± 1.0% and 6.9 ± 0.7 %, respectively) (Fig. 4d).

To determine the frequencies of deletions between two DSBs generated by a pegRNA and sgRNA, we checked aligned sequences that contained deletions with a length that was ±10 bp of the distance between the two potential DSBs. The total deletion frequency in the FANCF-targeted sample (48 ± 10 bp deletion) was 2.41 ± 0.12% for PE3 (H840A); the frequency was reduced to 1.72 ± 0.13%, 0.12 ± 0.04%, and 0.01 ± 0.01% for PE3 (H840A + N863A), PE3 (H840A + N854A), and PE3 (H840A + N863A + N854A), respectively (Fig. S5A). The HEK3-targeted sample (90 ± 10 bp deletion) had an average deletion frequency of 2.39 ± 0.17% for PE3 (H840A), which was reduced to 1.44 ± 0.08%, 0.08 ± 0.02%, and 0.04 ± 0.02% for PE3 (H840A + N863A), PE3 (H840A + N854A), and PE3 (H840A + N863A + N854A), respectively (Fig. S5B). Finally, the RUNX1-targeted sample (38 ± 10 bp deletion) had an average total deletion frequency of 0.50 ± 0.02%, 0.27 ± 0.02%, 0.01 ± 0.01%, and 0.01 ± 0.01% for PE3 (H840A), PE3 (H840A + N863A), PE3 (H840A + N854A), and PE3 (H840A + N863A + N854A), respectively (Fig. S5C). Based on our data, we could validate deletions between two gRNA-targeted sites generated by PE3 variants.

To examine the purity of correctly edited sequences, we calculated relative editing purity ratios [the frequency of correct edits normalized to that induced by PE3 (H840A) / the frequency of unwanted indels normalized to that induced by PE3 (H840A)] and the average editing purity [(the number of sequencing reads containing the correct edit) / (the number of sequencing reads containing the correct edit + the number of sequencing reads containing unwanted indels) * 100]. PE3 (H840A + N863A), PE3 (H840A + N854A) and PE3 (H840A + N854A + N863A) increased the relative editing purity ratios by 1.8-, 9.5-, and 9.4- folds respectively (Fig. 4e). The average editing purity was highest for PE3 (H840A + N854A, 95%) compared to PE3 (H840A, 86%), followed by PE3 (H840A + N854A + N863A, 95%) and PE3 (H840A + N863A, 90%) (Fig. 4f).

Collectively, our data show that PE (H840A + N863A) can significantly reduce the rate of unwanted indels in both the PE2 and

PE3 systems without sacrificing intended prime editing. Furthermore, PE3 (H840A + N854A) and PE3 (H840A + N854A + N863A) exhibited improved purity of intended editing, albeit with a lower range of intended edit frequencies than PE3 (H840A).

## Prime editing with engineered pegRNAs

As an additional means of increasing the frequency of desired edits, we incorporated epegRNAs into the PE3 system (ePE3). epegRNAs were developed by adding a structured RNA motif, such as evopreQ1 or mpknot, to the 3' end of the PBS sequence in pegRNAs. These RNA motifs, which are derived from virus sequences, protect pegRNAs from degradation and, thereby, improve both pegRNA stability and prime editing efficiency[27]. To test whether our PE variants can be applied to the epegRNA strategy, we assessed the efficiency of installing sub-stitutions, insertions, and deletions by PE variants together with epegRNAs.

First, we generated epegRNAs encoding different single-nucleotide substitutions at nine different genomic loci (FANCF, HEK3, EMX1, HEXA, PRNP, RNF2, RUNX1, VEGFA, and HBB sites (Fig. 5a and S6A)). Because PE (H840A + N863A) and PE (H840A + N854A) induced the highest and second highest frequencies of correct edits in both the PE2 and PE3 systems, we tested these variants in combination with epegRNAs. The frequencies of correct substitutions induced by ePE3 (H840A + N863A) (average, 33 ± 2%) were equal to those induced by ePE3 (H840A) (average, 32 ± 2%), whereas the average frequencies of unwanted indels were significantly decreased from 10 ± 1% for ePE3 (H840A) to 7 ± 0.9% for ePE3 (H840A + N863A) (Fig. S7A). Further-more, the frequencies of correct substitutions induced by ePE3 (H840A + N854A) (average, 18 ± 1.5%) were more than half that induced by ePE3 (H840A) with minimal unwanted indels (average, 0.67 ± 0.11%) (Fig. S7A). The purity of correct substitutions induced by ePE3 (H840A + N854A) was 14.5- and 8.5- fold higher than that for ePE3 (H840A) for editing at the FANCF and other sites, respectively (Fig. 5b and S6B). The purity of editing with ePE3 (H840A + N854A) reached 95.7% and 96.5% at the FANCF and other sites, respectively (Fig. 5c and S6C).

For further evaluation, we tested an epegRNA encoding a 24-bp Flag-tag insertion at five different genomic loci (HEK3, VEGFA, FANCF, RUNX1 and RNF2). When ePE3 (H840A + N863A) was used, the average frequency of correct insertions (17.8 ± 1.7%) was not significantly different from that induced by ePE3 (H840A) (19.3 ± 1.7%) at all five loci, whereas the average frequency of unwanted indels was significantly lower at 3.5 ± 0.7%, compared to that of ePE3 (H840A) (6.2 ± 1.2%) (Fig. 5d and S7B). Thus, the average purity of editing by ePE3 (H840A + N863A) was as high as 86.0%, whereas that of ePE3 (H840A) was 79.4% (Fig. 5f). In addition, the average frequency of correct insertions induced by ePE3 (H840A + N854A) was 6.8 ± 1%. Remark-ably, the average frequencies of unwanted indels induced by ePE3 (H840A + N854A) was 0.61 ± 0.11% (Fig. S7B). The relative editing pur-ity ratio for ePE3 (H840A + N854A) was 3.5-fold higher than that of ePE3 (H840A) (Fig. 5e). The highest editing purity (91%) for inserting the Flag-tag was achieved by ePE3 (H840A + N854A) (Fig. 5f).

Finally, we tested epegRNAs encoding a 15-bp deletion at five different loci (HEK3, VEGFA, FANCF, RUNX1 and RNF2) (Fig. 5g). Similar to results from the substitution and insertion experiments, ePE3 (H840A) and ePE3 (H840A + N863A) induced the same frequencies of the correct deletion (on average, 49.6 ± 5.0% and 48.4 ± 5.4%, respec-tively), but the frequency of unwanted indels induced by ePE3 (H840A + N863A) showed a trend to decrease at the five loci (Fig. 5g and S7C). In addition, when PE3 (H840A + N854A) combined with epegRNA was tested, the frequency of correct deletions reached 26.5 ± 4.9%, on average, but the average frequency of unwanted indels was 1.1 ± 0.3%, which led to an increase in the relative editing purity ratio, such that it was up to 5.0-fold higher than that of ePE3 (H840A) (Fig. 5h); the average editing purity for the deletion induced by ePE3

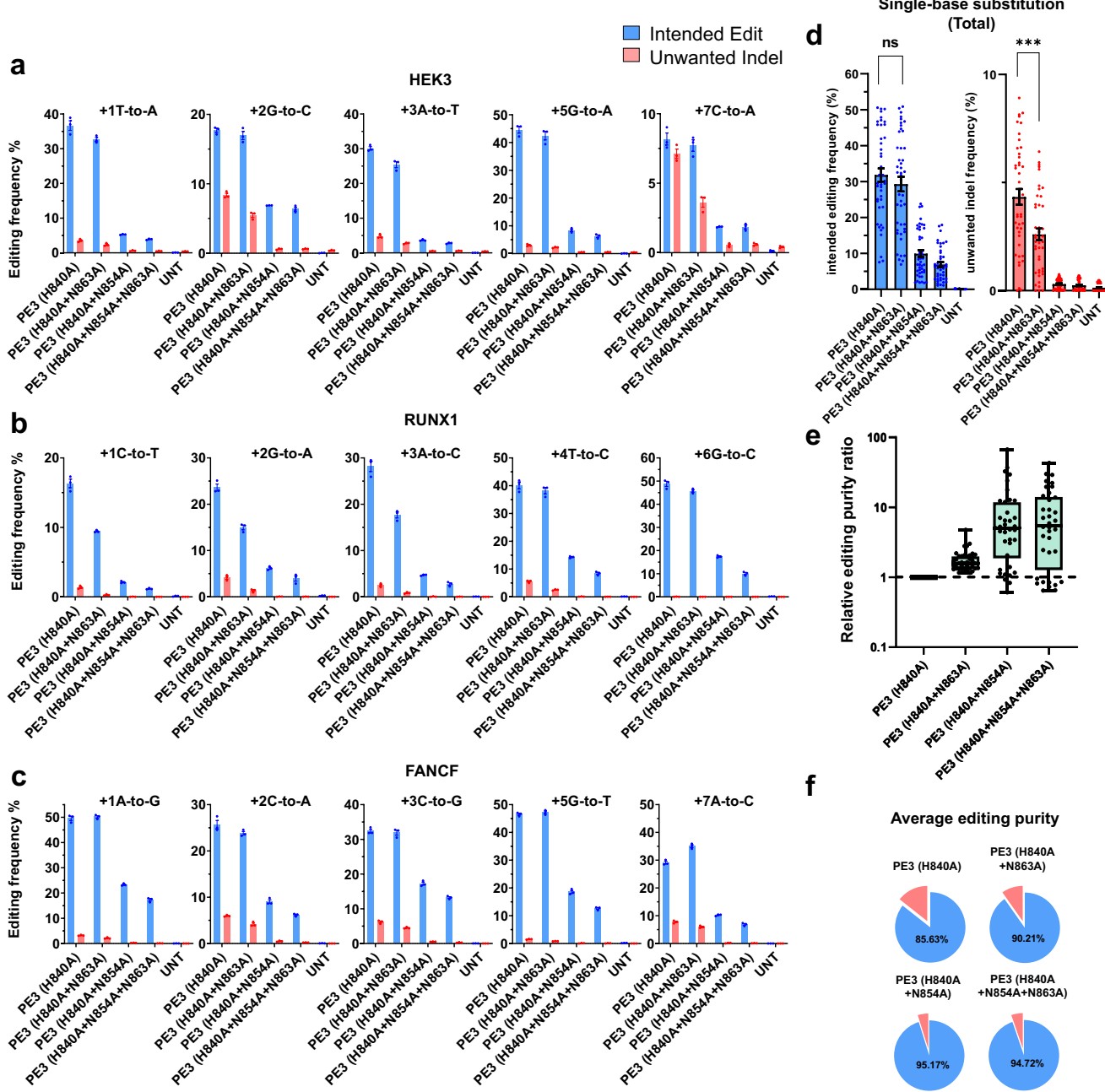

**Fig. 4 | PE variants that incorporate improved nCas9 variants, used in the PE3 system, induce fewer unwanted indels. a–c** Plasmids encoding PE (nCas9 variants), individual pegRNAs encoding five different single-base substitutions targeted to three sites, and nicking sgRNAs were transfected into HEK293T cells. The frequencies of the correct edits and unwanted indels at the *HEK3* (**a**), *RUNX1* (**b**), and *FANCF* (**c**) sites. The intended base substitutions are indicated at the top of each graph. **d** The average frequencies of the correct edits and unwanted indels for the five single-base substitutions at the *HEK3*, *RUNX1*, and *FANCF* sites. ns; $p = 0.355739$, ***$p = 0.000237$. **e**, **f** Relative editing purity ratios (the frequency of correct edits normalized to that induced by PE3 (H840A) / the frequency of

unwanted indels normalized to that induced by PE3 (H840A)) for the total single-base substitutions (**e**) and average editing purity (**f**) were calculated. Mean ± SEM (**a–c**) were determined three independent experiments. Mean ± SEM (**d**, **e**) of all individual values of sets of $n = 3$ independent replicates are shown. All statistical analysis for samples were conducted using unpaired Student's *t*-test (two-tailed) in GraphPad Prism 8. (ns, not significant, *$p < 0.05$, **$p < 0.01$, and ***$p < 0.001$ by student's *t*-test). For the boxes (**e**), the top, middle, and bottom lines represent the 25th, 50th, and 75th percentiles, respectively. The whiskers indicate min to max values. Source data are provided as a Source Data file.

(H840A + N854A) was 96%, whereas that of ePE3 (H840A) was 87% (Fig. 5i).

Next, to examine the editing efficiencies of the developed PE variants further, we tested them in combination with epegRNAs in additional cell lines including K562 (Fig. S8A and S8B) and HeLa (Fig. S9A and S9B). The relative editing purity induced by the ePE3 variants was increased up to 2.61- and 4.16-fold for ePE3 (H840A +

N863A) and ePE3 (H840A + N854A), respectively, compared to that of ePE3 (H840A) in K562 cells (Fig. S8C). The average editing purity was increased to 81.71% for ePE3 (H840A + N863A) and 83.40% for ePE3 (H840A + N854A), whereas that of ePE3 (H840A) was 79.35% (Fig. S8D). In HeLa cells, the relative editing purity induced by the ePE3 variants was increased up to 1.79- and 2.07-fold for ePE3 (H840A + N863A) and ePE3 (H840A + N854A), respectively, compared to that of ePE3

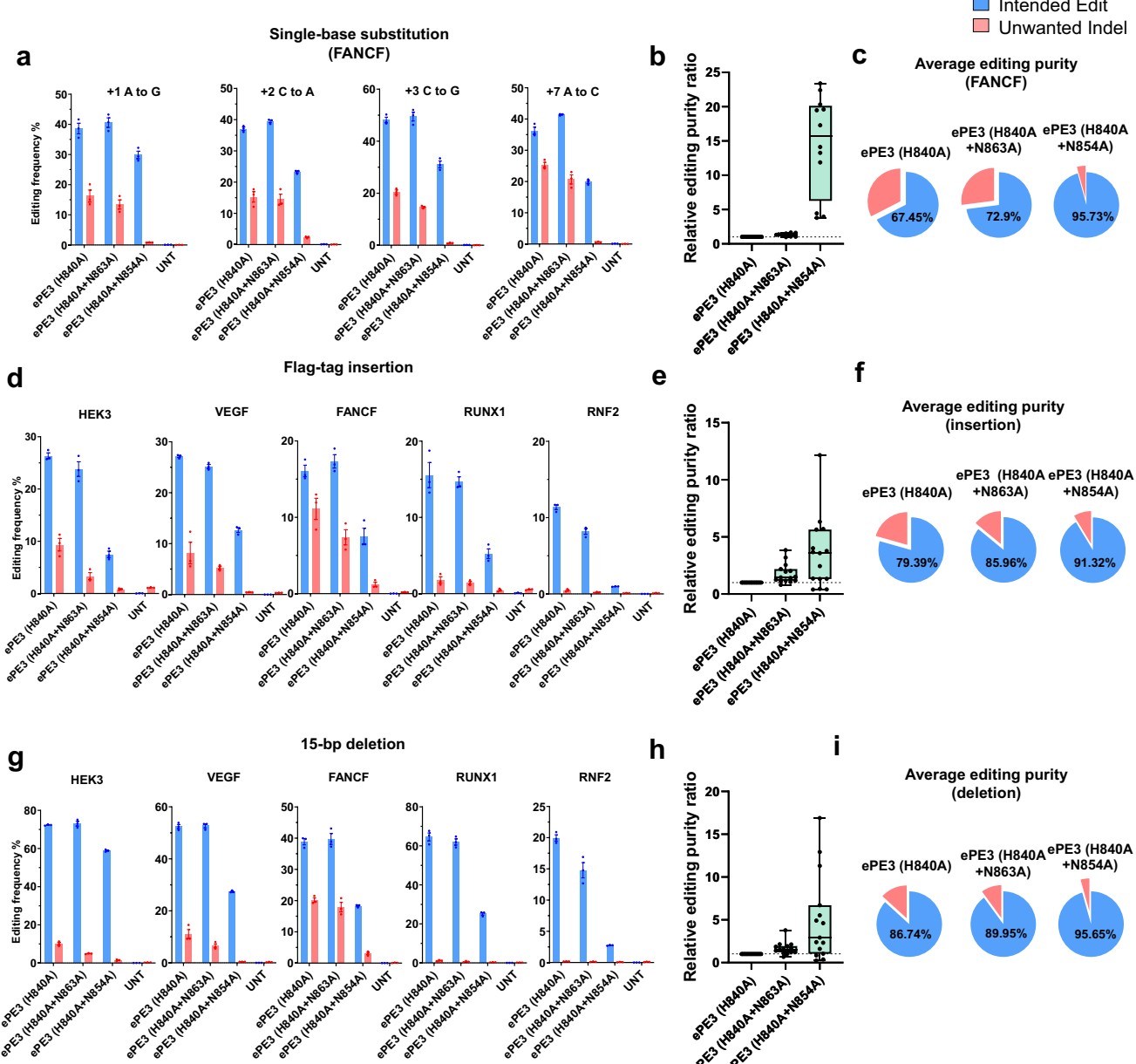

**Fig. 5 | epegRNAs used together with PE variants that incorporate nCas9 (H840A + N863A) and nCas9 (H840A + N854A) increase the purity of the correct edit for the PE3 system. a** Plasmids encoding PE (nCas9 variants), epegRNAs encoding four different single-base substitutions targeted to the *FANCF* site, and nicking sgRNAs were transfected into HEK293T cells. **b** Relative editing purity ratios (the frequency of correct edits normalized to that induced by ePE3 (H840A) / the frequency of unwanted indels normalized to that induced by ePE3 (H840A)) for the four single-base substitutions. **c**, The average editing purities associated with the PE (nCas9 variants) shown in pie charts. **d–i** Plasmids encoding PE (nCas9 variants), epegRNAs encoding 24-bp Flag-tag insertions (**d**) or 15-bp deletions (**g**) targeted to five different sites (*HEK3*, *VEGFA*, *FANCF*, *RUNX1*, and *RNF2*), and nicking sgRNAs

were transfected into HEK293T cells. Relative editing purity ratios normalized to the ePE3 (H840A) activity for insertions (**e**) and deletions (**h**). The average editing purities associated with the PE (nCas9 variants) for correct insertions (**f**) and deletions (**i**) shown in pie charts. Blue: intended edits; red: unwanted indels. Average editing purity: the number of reads containing the correct edit / the total number of reads containing edits (correct edits + unwanted edits) * 100. Mean ± SEM (**a**, **d**, **g**) were determined three independent experiments. Mean ± SEM (**b**, **e**, **h**) of all individual values of sets of *n* = 3 independent replicates are shown. For the boxes (**b**, **e**, **h**), the top, middle, and bottom lines represent the 25th, 50th, and 75th percentiles, respectively. The whiskers indicate min to max values. Source data are provided as a Source Data file.

(H840A) (Fig. S9C). The average editing purity was increased to as high as 83.32% for ePE3 (H840A + N854A) and 82.61% for ePE3 (H840A + N863A), whereas that of ePE3 (H840A) was 77.73% (Fig. S9D). Taken together, these results show that our PE variants led to similar editing outcomes in HEK293T, HeLa, and K562 cell lines.

In general, prime editing with epegRNAs increased the frequencies of the correct edit as well as that of unwanted indels. To decrease the frequency of such indels, we tested ePE3 containing nCas9 variants. When PE3 (H840A + N863A) was used with epegRNAs,

the frequency of correct edits was the same as that induced by ePE3 (H840A), but the frequency of unwanted indels was significantly reduced for substitutions and insertions, leading to a higher editing purity than that obtained for ePE3 (H840A). In addition, when the ePE3 (H840A + N854A) variant was used, the frequencies of correct edits were dramatically increased compared to those induced by PE3 (H840A + N854A), which induced relatively low editing frequencies. However, surprisingly, unlike the frequency of the correct edit, the frequency of unwanted indels was not increased even when epegRNAs

were used. Thus, the highest average editing purity, up to 96.5%, and up to a 14.5-fold higher relative editing purity ratio, was achieved by ePE3 (H840A + N854A) for all substitutions, insertions, and deletions.

## Discussion

In this study, we demonstrated that nCas9 (H840A) is not a bona fide nickase, often producing DSBs. Genomic DNA digestion in vitro with purified Cas9, nCas9 (D10A), or nCas9 (H840A), together with in vitro transcribed sgRNA, showed that Cas9 and nCas9 (D10A) generate DNA DSBs and nicks in the target strand, respectively, whereas nCas9 (H840A) can generate DSBs as well as the expected nicks in the non-target strand. To reduce this DSB-generating activity, we added the N863A mutation to nCas9 (H840A), which eliminated DSB formation at on-target sites. In addition, we found that the average number of genome-wide DSBs captured by Digenome-seq in nCas9 (H840A + N863A)-treated genomic DNA was significantly reduced compared to that in nCas9 (H840A)-treated genomic DNA. These results support the idea that nCas9 (H840A + N863A) also lacks the ability to generate off-target DSBs. To engineer nCas9 (H840A) further, potential catalytic amino acids (D839 and N854) were selected based on structural models and mutated to alanine. As expected, nCas9 (H840A + N863A) induced significantly lower indel frequencies than nCas9 (H840A) at all 15 tested loci in HEK293T cells. Interestingly, we found that other variants including nCas9 (H840A + N854A) and nCas9 (H840A + N854A + N863A) induced indel frequencies that were even lower than those induced by nCas9 (H840A + N863A).

Because PE consists of nCas9 (H840A) and M-MLV RT, we also incorporated these nCas9 variants into PE. We hypothesized that the unwanted indels generated by PE are caused the DSB-inducing activity of nCas9 (H840A). If true, the incorporation of nCas9 variants that lack such activity into PE should eliminate a high proportion of unwanted indels. We tested such nCas9 variants in the PE2 and PE3 systems. As expected, PE2 (H840A + N863A) significantly reduced unwanted indel production compared to PE2 (H840A). To increase the editing efficiency, we incorporated the nCas9 variants into the PE3 system. Because the PE3 system uses two guide RNAs, a pegRNA and a nicking sgRNA, we reasoned that two adjacent DSBs could be generated, potentially resulting in higher frequencies of unwanted indels compared with that seen with the PE2 system. For PE3 (H840A), both correct editing and unwanted indel frequencies were higher than observed for PE2 (H840A). However, when PE3 (H840A + N863A) was used instead, the correct editing frequency was the same as that of PE3 (H840A), but the frequency of unwanted indels was significantly decreased.

To increase the frequency of correct edits further, we used epegRNAs with the PE3 system. epegRNAs increase the frequency of correct edits induced by prime editing, but, problematically, they also increase the frequency of unwanted indels. To reduce the frequency of such indels, we tested PE3 variants together with epegRNAs. As anticipated, ePE3 (H840A + N863A) generated a reduced frequency of unwanted indels, but a similar frequency of correct edits, compared with ePE3 (H840A), for all substitutions, insertions, and deletions that we tested. In addition, ePE3 (H840A + N854A) induced a robustly decreased frequency of unwanted indels, despite the lower frequency of correct edits than ePE3 (H840A). Therefore, we highlight the finding that the relative editing purity ratio, normalized to that of ePE3 (H840A), was found to increase up to 14.5-fold for ePE3 (H840A + N854A), which was also associated with the highest average purity of editing (up to 96.48%).

In summary, we found that nCas9 (H840A) is not a bona fide nicakse, often generating DSBs in addition to nicks. With structure-guided mutagenesis, we were able to obtain bona fide nCas9 nickases that solely nick the non-target strand. By incorporating these nCas9 variants, we produced improved versions of PE that reduced or avoided unwanted indels for all of the prime editing systems that we

tested: PE2, PE3, and PE3 together with epegRNAs. Recently, another structurally-engineered pegRNAs have been reported to induce more efficient prime editing than the original pegRNA[28]. Combination of our improved versions of PE with these pegRNAs may increase editing efficiency and purity further.

Therapeutic genome editing requires not only a high editing frequency, but also very few editing byproducts, because minor unwanted indels may result in unexpected and adverse side effects. Therefore, efforts to improve the purity of the desired editing product are an important step for moving PEs toward clinical applications. The improved versions of PE developed here may represent a useful advance for therapeutic genome editing.

## Methods

### Plasmid construction for mammalian cell experiments

pCMV-PE2 (Addgene, #132775) was modified to incorporate mutations in the Cas9 domain (different combinations of the D839A, H840A, N854A, and N863A mutations or HNH-deleted mutations (residues 792-897, 765-908, 786-885, 824-874)) using gibson assembly (NEB-uilder HiFi DNA Assembly Cloning Kit, NEB #E5520). Sequences of Cas9 or Prime Editor plasmids used in study are listed in Supplementary Data 1. sgRNA-encoding plasmids were constructed by ligation (Quick Ligation Kit, NEB #M2200) of annealed oligonucleotides to pRG2 (Addgene, #104174) digested with BsaI (NEB, #R3733). pegRNA-encoding plasmids were constructed by Golden Gate assembly (NEB #E1601) using pU6-pegRNA-GG-acceptor (Plasmid #132777). Sequence of sgRNA and pegRNA constructs used in this study are listed in Supplementary Data 2–4.

### Protein purification

The plasmid encoding the His6-nCas9 (H840A + N863A) protein was generated by site-directed mutagenesis. Rosetta expression cells (EMD Millipore) were transformed with His6-Cas9-, His6-nCas9 (H840A)-, and His6-nCas9 (H840A + N863A)-expressing plasmids. Selected transformants were then cultured overnight in Luria-Bertani (LB) broth containing 100 μg/ml kanamycin at 37 °C. 10 ml overnight cultures of the cells were inoculated into 400 ml LB broth containing 100 μg/ml kanamycin, and cultured at 30 °C until the OD600 reached 0.5–0.6. Cell cultures were cooled to 16 °C for 1 h, supplemented with 0.5 mM IPTG, and cultured for 14-18 h. For protein purification, cells were harvested by centrifugation at 5000 g for 10 min at 4 °C and lysed by sonication in 5 ml lysis buffer (50 mM $NaH_2PO_4$, 300 mM NaCl, 1 mM dithiothreitol (DTT), and 10 mM imidazole, pH 8.0) supplemented with lysozyme (Sigma) and protease inhibitor (Roche complete, EDTA-free). The soluble lysate obtained after centrifugation at 18,000 g for 30 min at 4 °C was incubated with Ni-NTA agarose resin (Qiagen) for 1 h at 4 °C. The lysate/Ni-NTA mixture was applied to a column and washed with a buffer (50 mM $NaH_2PO_4$, 300 mM NaCl, and 20 mM imidazole, pH 8.0). The Cas9, nCas9 (H840A), and nCas9 (H840A + N863A) proteins were eluted with elution buffer (50 mM $NaH_2PO_4$, 300 mM NaCl, and 250 mM imidazole, pH 8.0). The buffer in the eluted protein solution was exchanged with storage buffer (20 mM HEPES-KOH (pH 7.5), 150 mM KCl, 1 mM DTT, and 20% glycerol); proteins were then concentrated with centrifugal filter units (Millipore).

### In vitro plasmid digestion and analysis

To generate control open-circular and linear plasmid forms, 1 μg of supercoiled plasmid was incubated with 2 units of Nt.BbvCI (NEB, #R0632S) or SpeI (NEB, #R3133S) at 37 °C for 2 h. To produce ribonucleoprotein complexes, purified Cas9, nCas9 (D10A), and nCas9 (H840A) (300 mM) were incubated with in vitro transcribed sgRNA (300 mM) for 15 min, then these ribonucleoprotein complexes were incubated with supercoiled plasmid for 8 h. After this incubation, the digested plasmids were cleaned with a PCR Product Purification Kit (MG Med, MK12020) and then subjected to agarose gel electrophoresis.

Band intensities of the supercoiled, open circular, and linearized plasmids were measured using ImageJ software.

### Whole genome sequencing and Digenome-seq
In vitro digested genomic DNA (1 μg) was fragmented to the 400- to 500-bp range using a Covaris system (Life Technologies) and blunt-ended using End Repair Mix (Thermo Fischer). Fragmented DNA was ligated with adapters to produce libraries, which were then subjected to WGS using a HiSeq X Ten Sequencer (Illumina) at Macrogen. WGS was performed at a sequencing depth of 30–40X. DNA cleavage sites were identified using the Digenome 2.0 program from our previous studies;[21] up-to-date versions of the program can be found at https://github.com/chizksh/digenome-toolkit2.

### Mammalian cell culture and transfection
HEK293T (ATCC CRL-11268) and HeLa (CCL-2) cells were maintained in Dulbecco's Modified Eagle Medium supplemented with 10% fetal bovine serum and 1% penicillin/streptomycin (Welgene). K562 (CCL-243) cells were maintained in RPMI 1640 (Welgene) with 10% fetal bovine serum and 1% penicillin/streptomycin (Welgene). Cells were not tested for mycoplasma contamination. HEK293T cells ($7.5 \times 10^4$) and HeLa cells ($3.5 \times 10^4$) were seeded in 48-well plates and transfected at ~80% confluency with Cas9 or nCas9 variant expression plasmids (750 ng) and an appropriate sgRNA expression plasmid (250 ng) using Lipofectamine 2000 (Invitrogen). Genomic DNA was isolated using a DNeasy Blood & Tissue Kit (Qiagen) at 72 h after transfection. For PE experiments, 800 ng PE expression vector, 200 ng pegRNA/epegRNA expression plasmid, and 83 ng nicking sgRNA expression plasmid were transfected using Lipofectamine 2000 (Invitrogen) according to the manufacturer's protocol. K562 cells were nucleofected using the SF Cell Line 4D-Nucleofector X Kit (Lonza) with $5 \times 10^5$ cells per sample (program FF-120), according to the manufacturer's protocol. 1500 ng prime editor expression plasmid, 500 ng pegRNA expression plasmid, and 150 ng nicking sgRNA expression plasmid were nucleofected in 100 μl nucelofector cuvettes. Genomic DNA was isolated using a DNeasy Blood & Tissue Kit (Qiagen) at 72 h after transfection.

### Targeted deep sequencing
To analyze the frequency of edits, on-target sites were first amplified via nested primary PCR, a secondary PCR, and a third PCR using TruSeq HT Dual index-containing primers and PrimeSTAR® GXL DNA Polymerase (TAKARA) to generate deep sequencing libraries. The libraries were sequenced using Illumina MiniSeq with paired-end sequencing systems. The prime editing and unwanted indel frequencies are presented as percentages of sequencing reads containing correct edits or indels among total sequencing reads. The computer program used to analyze the frequency of edits is available at https://github.com/ibs-cge2/prime_editor_analysis. The PCR primer sequences are shown in Supplementary Data 5.

### Data analysis
For data analysis and visualization, we used IGV (2.5.3), Microsoft Excel (2016), Powerpoint (2016), BWA (v.0.7.17), SAMtools (v.1.9), and Prism 8 (8.4.3) for drawing figures, graphs, and tables. For the structural assay, we used PyMol (2.5.2)[29].

### Statistics and reproducibility
Sample sizes were determined based on previous publications of our and other groups generating reproducible results for genome editing experiments. Data are presented as means ± SEM from independent experiments. P-values were calculated by two-tailed, unpaired Student's t-test. The individual evaluation experiments of HEK293T cells, K562 cells, and HeLa cells were independently repeated three times, with comparable results. No data was excluded from the analyses.

Samples were not randomized. Investigators were not blinded during experiments and data analysis.

### Reporting summary
Further information on research design is available in the Nature Portfolio Reporting Summary linked to this article.

## Data availability
DNA sequencing data have been deposited in the National Center for Biotechnology Information (NCBI) Sequence Read Archive (SRA) database with BioProject accession code PRJNA888859 . The protein-structure data for HNH domain of Cas9 is from Protein Data Bank (https://www.rcsb.org) with PDB code 6OOY[25]. Source data are provided with this paper.

## Code availability
Correct editing frequencies and unwanted indel frequencies from targeted deep sequencing data were calculated with source code written by Python (version 3.9) (https://github.com/ibs-cge2/prime_editor_analysis, written by BotBot Inc.). DOI: 10.5281/zenodo.7726909. We used the 'maund_default.py' script to analyze base substitution frequencies and unwanted indel frequencies. For calculating the correct editing frequencies from insertion and deletion PE experiments, we used the 'prime_editor.py' script with default parameters. The sequence information for the input ('aseq' and 'rgen' for maund_default.py, and '–target_seq' and 'amplicon_seq' for prime_editor.py) is listed in Supplementary Data 5. For more details, please refer to the 'README.md' file on the website.

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

## Acknowledgements
This work was supported by Institute for Basic Science (IBS-R021-D1 to J.-S.K).

## Author contributions
J.-S.K. supervised the research. J.-S.K., J.L., and K.Y. wrote the paper. J.L. and K.L. designed the experiments. J.L., K.L., A.K., Y.G.M., E.C., S.-I.C., and J.M.L. performed the experiments and bioinformatics analysis. All authors discussed the results and commented on the manuscript.

## Competing interests
J.-S.K., K.Y., and J.L. have published patent (no. WO2021215897A1) related to this study. (Applicant: Institute for Basic Science; Inventor: Jin-Soo Kim, Kayeong Lim and Jaesuk Lee; Status: Published). The other authors declare no competing interests.
