## [Peer Review File · Nature Communications]

Reviewers' Comments:

Reviewer #1:

Remarks to the Author:

In this manuscript the authors reported that nCas9 (H840A) can cleave both DNA strands at low frequency compared to wild-type Cas9. This produces unwanted DSBs and indels. Double mutant nCas9 in the HNH nuclease domain did not induce DSB in vitro and reduced unwanted indels compared to H840A. The authors further showed that the nCas9 variant (H840A+N854A) in prime editor and with engineered pegRNAs increased the ratio of correct edits, but not unwanted indels. The finding of unexpected DSB induced by nCas9 (H840A) is an important contribution to the prime editing field. Overall this is a significant manuscript which will improve the PE editing purity. Several questions need to be addressed in the revision.

Major:

1. The statement of "Indel-free prime editing" is not accurate. The authors showed that in the PE3 setting, H840A+N854A and H840A+N854A+N863A still had 4-5% indels out of overall editing (Fig. 4f). Although the indel rate is reduced or minimized, it is still detectable.
2. Fig. 2 showed H840A-induced DSB sites detected by Digenome-seq. Cas9 can induce indels at off-target sites. Does H840A or PE2 also induce low-level indels at some off-target sites?

Minor:

- Line 488: "HEK29T cells" should be HEK293T.

Reviewer #2:

Remarks to the Author:

In this manuscript, Lee et al. found nCas9 (H840A) but not nCas9 (D10A) can cleave both DNA strands. The authors also found that the double-mutant nCas9 (H840A+N863A) did not exhibit the DSB-inducing behavior in vitro. Either alone or in fusion with the M-MLV reverse transcriptase (prime editor), the double-mutant nCas9 induced a lower frequency of unwanted indels, compared to H840A. When incorporated into prime editor and used with engineered pegRNAs, the authors found that the nCas9 variant (H840A+N854A) dramatically increased the frequency of correct edits, but not unwanted indels, yielding the highest purity of editing outcomes compared to H840A. This study is of potential interest to the gene editing field and the following two points are suggested to revise the manuscript.

1. In this study, the authors chose D839A, N854A and N863A to further reduce the unwanted indels. According to a previously published study (Fonfara et al. 2014, Nucleic Acids Res), D986, H983, E762 can also be mutated to induce pure single strand break. The authors are suggested to generate more Cas9 variants and test their single-strand DNA cleavage activities.
2. To further prove the efficacy of the newly-developed PE variants, it is suggested to test them at more genomic sites and in different cell lines.

Reviewer #3:

Remarks to the Author:

In the present manuscript titled "Indel-free prime editing with bona fide Cas9 nickases", the authors reported that nCas9 (H840A) can cleave both strands which less efficiently than wild-type Cas9. With structure-guided mutagenesis, they also obtained nCas9 nickases that solely nick the non-target strand, incorporating these nCas9 variants with prime editing systems, they reduced or avoided unwanted indels. Overall, this study is meaningful, especially the discovery of nCas9 (H840A) has a little double strand breaking activity. I only have several minor comments:

1. The title "Indel-free" is not accurate, in most cases they just decrease the indel frequency.
2. At 15 endogenous genomic sites, they found that additional mutations in the HNH domain further reduce indel formation. To be clear, additional mutations in the HNH may also reduce the overall performance, such as binding activity or non-target strand breaking activity, which may

also decrease indel efficiency. The authors need to explain the problem.

3.Line 336-338, "Because the PE3 system 337 uses two guide RNAs, a pegRNA and a nicking sgRNA, we reasoned that two adjacent DSBs 338 could be generated", the reviewer suggests that the authors should provide the deletion efficiency between the two DSBs for PE3 variants at Figure 4.

Point-by-point response

We would like to thank the three reviewers for their helpful comments. To address the issues raised by the reviewers, we have now made appropriate changes in our revised manuscript. Added or edited sentences are underlined in the main text for clarity.

Reviewer #1 (Remarks to the Author):

In this manuscript the authors reported that nCas9 (H840A) can cleave both DNA strands at low frequency compared to wild-type Cas9. This produces unwanted DSBs and indels. Double mutant nCas9 in the HNH nuclease domain did not induce DSB in vitro and reduced unwanted indels compared to H840A. The authors further showed that the nCas9 variant (H840A+N854A) in prime editor and with engineered pegRNAs increased the ratio of correct edits, but not unwanted indels. The finding of unexpected DSB induced by nCas9 (H840A) is an important contribution to the prime editing field. Overall this is a significant manuscript which will improve the PE editing purity. Several questions need to be addressed in the revision.

Major:

1. The statement of “Indel-free prime editing” is not accurate. The authors showed that in the PE3 setting, H840A+N854A and H840A+N854A+N863A still had 4-5% indels out of overall editing (Fig. 4f). Although the indel rate is reduced or minimized, it is still detectable.

Response: This reviewer is right! We have now changed the title as follows: Prime editing with bona fide Cas9 nickases minimizes unwanted indels.

2. Fig. 2 showed H840A-induced DSB sites detected by Digenome-seq. Cas9 can induce indels at off-target sites. Does H840A or PE2 also induce low-level indels at some off-target sites?

Response: To detect any off-target indels induced by nCas9 (H840A), nCas9 (H840A+N863A), PE2 (H840A), and PE2 (H840A+N863A), we have performed targeted deep sequencing at potential off-target sites in *RUNX1*- and *EMX1*-targeted samples. We were able to detect low frequencies of indels at some off-target sites. We have now added Supplementary Figure 2 and the following paragraph in the Results section on p.7-8 as follows: “To examine whether off-target indels were induced by nCas9 (H840A), nCas9 (H840A+N863A), and WT Cas9, we examined indel frequencies in *RUNX1*- and

EMX1-targeted samples at candidate off-target sites. DSB sites detected by nCas9 (H840A)-induced Digenome-seq (Figure 2B) were used for validation. The Digenome score (≥ 8.0) and the number of mismatched bases (≤ 6 bp) were applied to filter candidate sites. Indel frequencies induced by nCas9 variants and WT Cas9 at off-target sites in the *RUNX1*- (Figure S2A) and *EMX1*- (Figure S2D) targeted samples were then measured. In addition, because prime editor uses nCas9 (H840A), we also examined whether off-target indels were generated by PE2 and PE2 (H840A+N863A). The same off-target candidate sites were tested for intended edits and unwanted indels in *RUNX1*- (Figure S2B and S2C) and *EMX1*- (Figure S2E and S2F) targeted samples. We were able to confirm the presence of low frequencies of indels induced by nCas9 (H840A) and PE2 at some off-target sites. Average indel frequencies of 0.027% and 0.035% were detected at the *RUNX1* off-2 site in nCas9 (H840A)- and PE2-treated samples, respectively.”

Supplementary Figure 2. Editing frequency of nCas9 variants and PE2 variants at off-target sites

Minor:

- Line 488: “HEK29T cells” should be HEK293T.

Response: We have now made appropriate changes.

Reviewer #2 (Remarks to the Author):

In this manuscript, Lee et al. found nCas9 (H840A) but not nCas9 (D10A) can cleave both DNA strands. The authors also found that the double-mutant nCas9 (H840A+N863A) did not exhibit the DSB-inducing behavior in vitro. Either alone or in fusion with the M-MLV reverse transcriptase (prime editor), the double-mutant nCas9 induced a lower frequency of unwanted indels, compared to H840A. When incorporated into prime editor and used with engineered pegRNAs, the authors found that the nCas9 variant (H840A+N854A) dramatically increased the frequency of correct edits, but not unwanted indels, yielding the highest purity of editing outcomes compared to H840A. This study is of potential interest to the gene editing field and the following two points are suggested to revise the manuscript.

1. In this study, the authors chose D839A, N854A and N863A to further reduce the unwanted indels. According to a previously published study (Fonfara et al. 2014, Nucleic Acids Res), D986, H983, E762 can also be mutated to induce pure single strand break. The authors are suggested to generate more Cas9 variants and test their single-strand DNA cleavage activities.

Response: Please note that D10 (RuvC I), D986 (RuvC III), H983 (RuvC III), and E762 (RuvC II) are amino-acid residues in the RuvC domain catalyzing non-target strand cleavage and that H840, D839, N854, and N863 are residues in the HNH domain catalyzing target strand cleavage. In this study, we sought to inactivate the HNH, rather than RuvC, catalytic activity by introducing additional mutations in the HNH domain.

2. To further prove the efficacy of the newly-developed PE variants, it is suggested to test them at more genomic sites and in different cell lines.

Response: To further examine the efficiencies of the newly-developed PE variants, we investigated their activities at five more genomic sites in HEK293T cells and in two additional cell lines (K562 and HeLa). We have now updated the editing outcomes in HEK293T cells to include the additional target sites in the Results section on p.13-14, in Figure 5, and in Supplementary Figure 6. In addition, the editing results in the additional cell lines have been added; the information about K562 cells is in Supplementary Figure 8 and that for HeLa cells is in Supplementary Figure 9. A description of the editing results for these additional cell lines has been added to the Results section on p.14 – 15, as follows:

“Next, to examine the editing efficiencies of the newly-developed PE variants further, we tested them in combination with epegRNAs in additional cell lines including K562 (Figure S8A and S8B) and HeLa (Figure S9A and S9B). The relative editing purity induced by the ePE3 variants was increased up to 2.61- and 4.16-fold for ePE3

(H840A+N863A) and ePE3 (H840A+N854A), respectively, compared to that of ePE3 (H840A) in K562 cells (Figure S8C). The average editing purity was increased to 81.71% for ePE3 (H840A+N863A) and 83.40% for ePE3 (H840A+N854A), whereas that of ePE3 (H840A) was 79.35% (Figure S8D). In HeLa cells, the relative editing purity induced by the ePE3 variants was increased up to 1.79- and 2.07-fold for ePE3 (H840A+N863A) and ePE3 (H840A+N854A), respectively, compared to that of ePE3 (H840A) (Figure S9C). The average editing purity was increased to as high as 83.32% for ePE3 (H840A+N854A) and 82.61% for ePE3 (H840A+N863A), whereas that of ePE3 (H840A) was 77.73% (Figure S9D). Taken together, these results show that our PE variants led to similar editing outcomes in HEK293T, HeLa, and K562 cell lines.”

Supplementary Figure 6. PE variants that incorporate improved nCas9 variants, used in the PE3 system with epegRNAs, achieve higher frequencies of substitutions

Supplementary Figure 8. epegRNAs used together with PE variants that incorporate nCas9 (H840A+N863A) or nCas9 (H840A+N854A) increase the purity of the correct edit for the PE3 system in K562 cells

Supplementary Figure 9. epegRNAs used together with PE variants that incorporate nCas9 (H840A+N863A) or nCas9 (H840A+N854A) increase the purity of the correct edit for the PE3 system in HeLa cells

Reviewer #3 (Remarks to the Author):

In the present manuscript titled “Indel-free prime editing with bona fide Cas9 nickases”, the authors reported that nCas9 (H840A) can cleave both strands which less efficiently than wild-type Cas9. With structure-guided mutagenesis, they also obtained nCas9 nickases that solely nick the non-target strand, incorporating these nCas9 variants with prime editing systems, they reduced or avoided unwanted indels. Overall, this study is meaningful, especially the discovery of nCas9 (H840A) has a little double strand breaking activity. I only have several minor comments:

1. The title “Indel-free” is not accurate, in most cases they just decrease the indel frequency.

Response: We have now changed the title as follows: Prime editing with bona fide Cas9 nickases minimizes unwanted indels.

2. At 15 endogenous genomic sites, they found that additional mutations in the HNH domain further reduce indel formation. To be clear, additional mutations in the HNH may also reduce the overall performance, such as binding activity or non-target strand breaking activity, which may also decrease indel efficiency. The authors need to explain the problem.

Response: We would like to thank this reviewer for carefully considering our manuscript. We have added the following explanation about this problem in Lines 183-187 as follows:

“However, these additional mutations in the HNH domain may also reduce its general functionality, such as its binding activity or protein folding, which may in turn reduce the efficiency of indel generation. To determine whether these nCas9 variants are still functional, we have incorporated them into the prime editor system.”

3. Line 336-338, “Because the PE3 system uses two guide RNAs, a pegRNA and a nicking sgRNA, we reasoned that two adjacent DSBs could be generated”, the reviewer suggests that the authors should provide the deletion efficiency between the two DSBs for PE3 variants at Figure 4.

Response: We appreciate this reviewer's thoughtful consideration of our work. We have checked the frequencies of deletions between the two gRNA (pegRNA and nicking sgRNA) targeted sites for PE3 variants and show the results in Figure 4. We have now added Supplementary Figure 5 and described the results on p.11 – 12 as follows:

“To determine the frequencies of deletions between two DSBs generated by a pegRNA

and sgRNA, we checked aligned sequences that contained deletions with a length that was ± 10 bp of the distance between the two potential DSBs. The total deletion frequency in the *FANCF*-targeted sample (48 ± 10 bp deletion) was $2.41 \pm 0.12\%$ for PE3 (H840A); the frequency was reduced to $1.72 \pm 0.13\%$, $0.12 \pm 0.04\%$, and $0.01 \pm 0.01\%$ for PE3 (H840A + N863A), PE3 (H840A + N854A), and PE3 (H840A + N863A + N854A), respectively (Figure S5A). The *HEK3*-targeted sample (90 ± 10 bp deletion) had an average deletion frequency of $2.39 \pm 0.17\%$ for PE3 (H840A), which was reduced to $1.44 \pm 0.08\%$, $0.08 \pm 0.02\%$, and $0.04 \pm 0.02\%$ for PE3 (H840A + N863A), PE3 (H840A + N854A), and PE3 (H840A + N863A + N854A), respectively (Figure S5B). Finally, the *RUNX1*-targeted sample (38 ± 10 bp deletion) had an average total deletion frequency of $0.50 \pm 0.02\%$, $0.27 \pm 0.02\%$, $0.01 \pm 0.01\%$, and $0.01 \pm 0.01\%$ for PE3 (H840A), PE3 (H840A + N863A), PE3 (H840A + N854A), and PE3 (H840A + N863A + N854A), respectively (Figure S5C). Based on our data, we could validate deletions between two gRNA-targeted sites generated by PE3 variants.”

Supplementary Figure 5. Deletion frequencies between two gRNA (pegRNA and nicking sgRNA) targeted sites.

Reviewers' Comments:

Reviewer #1:

Remarks to the Author:

The authors have significantly improved the manuscript in the revision. All my questions have been adequately addressed.

Reviewer #2:

Remarks to the Author:

In this manuscript, the authors have revised the manuscript thoroughly and my previous concerns have been fully addressed. Just one minor point is suggested and I recommend to accept the revised manuscript.

Minor point

The authors showed their engineered nickase, when used with epegRNA, can yield higher product purity. Recently, another structurally-engineered pegRNA (apegRNA, PMID: 35351879) has been reported to induce more efficient prime editing than the original pegRNA. The authors are suggested to discuss the possibility to combine their engineered nickase with apegRNA to achieve higher editing frequency or product purity.

Reviewer #3:

Remarks to the Author:

The author has answered my question and I have no further questions.

Point-by-point response

We would like to thank the three reviewers for their helpful comments. To address the issues raised by the reviewers, we have now made appropriate changes in our revised manuscript. Added or edited sentences are underlined in the main text for clarity.

Reviewer #1 (Remarks to the Author):

The authors have significantly improved the manuscript in the revision. All my questions have been adequately addressed.

Response: We would like to appreciate this reviewer for helpful comments.

Reviewer #2 (Remarks to the Author):

In this manuscript, the authors have revised the manuscript thoroughly and my previous concerns have been fully addressed. Just one minor point is suggested and I recommend to accept the revised manuscript.

Response: We would like to thank this reviewer for helpful comments.

Minor point

The authors showed their engineered nickase, when used with epegRNA, can yield higher product purity. Recently, another structurally-engineered pegRNA (apegRNA, PMID: 35351879) has been reported to induce more efficient prime editing than the original pegRNA. The authors are suggested to discuss the possibility to combine their engineered nickase with apegRNA to achieve higher editing frequency or product purity.

Response: We have now mentioned the possibility in Discussion section line:402-404 as follows:

“Recently, another structurally-engineered pegRNAs have been reported to induce more efficient prime editing than the original pegRNA. Combination of our improved versions of PE with these pegRNAs may increase editing efficiency and purity further.”

Reviewer #3 (Remarks to the Author):

The author has answered my question and I have no further questions.

Response: We would like to thank this reviewer for helpful comments.